# Simple Process for Flexible Light-Extracting QD Film and White OLED

**DOI:** 10.3390/mi16121367

**Published:** 2025-11-30

**Authors:** Eun Jeong Bae, Tae Jeong Hwang, Geun Su Choi, Yong-Min Lee, Byeong-Kwon Ju, Young Wook Park, Dong-Hyun Baek

**Affiliations:** 1Display and Nanosensor Laboratory, Department of Electrical Engineering, Korea University, 145, Anam-ro, Seongbuk-gu, Seoul 02841, Republic of Korea; baeejng@korea.ac.kr (E.J.B.); bkju@korea.ac.kr (B.-K.J.); 2Nano and Organic-Electronics Laboratory, Department of Display and Semiconductor Engineering, Sunmoon University, Asan 31460, Chungcheongnam-do, Republic of Korea; zeratull1234@sunmoon.ac.kr; 3Research Center for Nano-Bio Science, Sunmoon University, Asan 31460, Chungcheongnam-do, Republic of Korea; ymlee@sunmoon.ac.kr; 4Center for Next Generation Semiconductor Technology, Department of Display and Semiconductor Engineering, Sunmoon University, Asan 31460, Chungcheongnam-do, Republic of Korea

**Keywords:** organic light-emitting diodes (OLEDs), quantum dot (QD), light extraction, flexible color conversion, white OLED

## Abstract

Quantum dots (QDs) have tremendous potential for next-generation displays due to their high color purity, photoluminescence efficiency, and power efficiency. In this work, we present a simple and cost-effective method for fabricating flexible single- and multiple-layer films, and they can be detached and attached to the outside of OLEDs as a light-scattering and color-conversion layer. Light extraction efficiency is enhanced by forming low-density structures by using the reactive ion etching (RIE) process. As a result, the QD/PDMS composite film allowed for color conversion and achieved an excellent light extraction efficiency of up to 9.2%. Furthermore, the QD/PDMS composite film and greenish-blue OLED produced white light (CIEx,y = 0.28, 0.41), demonstrating the potential for application in broad areas, from flexible displays to lighting. The method provides a simple and cost-effective alternative to conventional processes.

## 1. Introduction

Quantum dots (QDs) are attractive materials due to their excellent color purity, high luminescence efficiency, energy efficiency, and controllable band gap characteristics as next-generation display devices. Furthermore, many researchers have been actively conducting studies on their excellent color reproducibility through fusion with organic light-emitting diodes (OLEDs) or light-emitting diodes (LEDs) [1,2,3,4,5].

High-energy blue light from an OLED or LED can be converted to green and red light in a QD. Therefore, blue light can generate red and green light with high color purity and expand the color gamut. Currently, QDs are replacing phosphors as the desired color-producing material to enhance the color performance of various displays. In previous studies, QD patterns have been used to define sub-pixels in inkjet printing, transfer printing, the photolithography process, and directly internal OLED structures and hybrid QLED devices with a combination of solution processing and conventional vacuum evaporation [6]. As mentioned, these methods require expensive and specific functional equipment, and they still have problems such as a complex process and limited production efficiency and yield. In addition, the low luminescence efficiency of QDs still remains [7,8]. For these reasons, researchers are increasingly developing simplified processes such as attaching or transferring QD layers to the outside of devices. On the other hand, QDs are a core technology that internally generates photon extraction from the inside to the outside of a device to increase the luminous efficiency. Light is generated within the device through the recombination of electrons and holes, but a significant portion of this light becomes trapped internally due to refractive index mismatches at the interfaces between the organic layer and substrate and between the substrate/air [9,10,11]. Overcoming this internal light loss is difficult through the luminescent properties of the materials, making structural approaches crucial for improving light extraction efficiency.

To address this problem, various nanostructure-based light extraction layers have been studied. However, most structures rely on complex processes or inflexible materials, limiting their applicability to large-area flexible displays. Recently, research on external light extraction structures that can be attached to the outside of devices has actively been conducted to solve drawbacks [12,13,14,15,16]. In particular, PDMS (polydimethylsiloxane) is a representative material that is transparent and flexible, has excellent formability, and is easy to mix with high-refractive index nanoparticles, making it highly useful, for which it stands out as external light extraction layer. PDMS has superior properties and forms micro/nano-structures through spin-coating, breath figure, drop casting and etching processes compared to conventional lithographic processes [17,18,19,20].

In this study, we propose a method for fabricating a flexible QD/PDMS composite film as a color conversion layer and for forming a micro/nano-texture with a surface etching process to enhance the external light extraction efficiency of OLEDs. To improve light scattering and extraction, we performed the reactive ion etching (RIE) process on the top surface of a QD/PDMS composite layer, which induced a micro/nano-structure with nanoscale width and microscale length. Additionally, to analyze other characteristics of the QD-applied structure, properties were analyzed and optimized based on the thickness of the adhesive film between the OLED glass and the QD dispersion layer. The proposed structure has advantages of simultaneously improving external light extraction efficiency and achieving color conversion without additional complex optical designs.

Herein, we present a simple and effective method for enhancing light extraction efficiency and color conversion by applying a micro-wrinkle structure to QD/PDMS composite film. The RIE process produced a micro/nano-structure on PDMS film, which was attached to a blue OLED for light extraction, achieving an External Quantum Efficiency (EQE) of 9.2%. Furthermore, we show that red light emitted in combination with the green and blue light from a green-blue OLED generates white light. This approach can provide solutions for various applications, such as display devices and lighting.

## 2. Materials and Methods

### 2.1. Fabrication of QD/PDMS and PDMS/QD/PDMS Composite Film

We applied the supporting layer with bare PDMS, which produced 10:1 (PDMS base: curing agent) of weight% mixed in a plastic cup. To ensure a uniform thickness, we spin-coated bare PDMS (20 μm) at 300 rpm on glass substrate. The following process was for the sample to be cured at 80 °C for 1 h to produce the supporting layer.

We used dispersed red-QD solution in hexane to make QD/PDMS composite film as the color conversion layer. A total of 1.1 wt% of QD solution was mixed with 1 g of PDMS base in the glass dish by mechanically mixing. After that, 0.3 g of PDMS curing agent was poured in a glass dish for 1 min mixing. The QD solution was spin-coated on the supporting layer at 1000, 2000, and 3000 rpm. And then this layer was cured at 80 °C for 1 h. Finally, we completed a QD/PDMS single-layer film. The multiple-layer film (PDMS/QD/PDMS) comprised one more bare PDMS layer, which was spin-coated on the QD/PDMS layer under equal conditions. So, we designed a top PDMS layer to cover the QD layer to keep its flexibility, as in Figure 1.

This process was designed to fully cover and protect the QD layer with PDMS while maintaining flexibility. The fabricated film formed a micro-nano wrinkle structure on the PDMS surface via RIE. The RIE process was conducted in two steps. First, we performed a pre-treatment with 20 sccm Argon (Ar) plasma at 150 W (working pressure of 32 mtorr) for 2.5 min in order to remove surface contaminants and activate the PDMS surface. The pre-treatment induced the activation of chemical bonds on the surface, enhancing its reactivity to the plasma for subsequent etching. In the second step, the etching process was set up with three types of substances: CF_4_, CF_4_ + O_2_ (4:1), and CHF_3_. The RF power, gas flow rate, and working pressure were equal to those in the first step, and the processing time was 7.5 min. During the plasma etching process, the PDMS surface was selectively etched, and residual stress induced the formation of micro-wrinkles. These wrinkle patterns formed differently depending on the gas composition, consequently affecting the light scattering characteristics. As shown in Figure 2, the processed QD/PDMS composite film could be detached from the substrate because the surface of fully cured QD/PDMS film had low surface energy, so the composite could be released from the surface with small amounts of physical force. However, the fabricated films were very thin (20 um), so it was easy for them to cling to each other. To solve the problem, KAPTON tape was attached to the four edges of the QD/PDMS film to support them so that it would prevent them from sticking to each other when separated from the substrate.

Resultingly, we utilized irreversible and reversible methods to attach and detach the QD/PDMS film and blue OLED to the exterior of a blue OLED device to evaluate its light extraction and color conversion properties.

### 2.2. Fabrication of OLED

A substrate with 185 nm ITO deposited on soda-lime glass was cleaned using an ultrasonic cleaner with acetone, methanol, and deionized water in sequence for 15 min each. The cleaned substrates were dried in a 120 °C dry oven for 1 h. The OLED light-emitting area was defined as a 6.25 mm diameter circle using photoresist (AZ 601 GXR, AZ Electronic Materials Co., Ltd.) during the photolithography process. The prepared substrates were treated with UV ozone (UVC-300, Omniscience) and O_2_ plasma (CUTE, Femto Science Co.) to remove residual contaminants and adjust the work function of the anode to lower the driving voltage. In this study, a phosphorescent OLED was fabricated. The device structures are as follows. ITO (185 nm)/Dipyrazino[2,3-f:2′,3′-h]quinoxaline-2,3,6,7,10,11-hexacarbonitrile (HAT-CN) (1 nm)/4,4-Cyclohexylidenebis[N,N-bis(4-methylphenyl)benzenamine] (TAPC) (56 nm)/Tris(4-carbazoyl-9-ylphenyl)amine (TCTA) (30 nm)/Bis[2-(4,6-difluorophenyl)pyridinato-C2,N](picolinato)iridium (Firpic) (0.15 nm)/1,3,5-Tris(3-pyridyl-3-phenyl)benzene (TmPyPB) (54 nm)/LiF (1.25 nm)/Al. All organic materials and metals used in this study were rotary evaporated in an approximately 10^−7^ torr high vacuum at 12 rpm, with the deposition rates of the organic material and metal controlled at 1 Å/s and 3 Å/s, respectively. The thickness of each thin film was controlled using a 6 MHz gold-coated quartz crystal microbalance (QCM, Phillips Technologies) and a PCI Express interface thin film deposition controller (IQM-233, INFICON). The fabricated devices were stored in a glove box maintained under an Ar environment with a moisture concentration below 1 ppm.

### 2.3. Characterization of the Fabricated OLEDs

The electroluminescence (EL) characteristics of the fabricated OLED devices were measured in a low-vacuum (10^−3^ torr), darkroom environment using a spectroradiometer (CS-2000A, Konica Minolta Co.) and a source meter (Keithley 2400, Tektronix). The samples were fixed in a vacuum holder to minimize the influence of the external environment during measurement. Furthermore, to evaluate the light extraction characteristics and viewing angle dependence of the OLEDs, the light emission characteristics were measured at vertical viewing angles ranging from 0 to 70° relative to the normal direction of the light-emitting surface. The EQE was recalculated based on the data obtained from the viewing angle measurements [21] (Appendix A).
$${EQE}_{recalculated}={CF\times EQE}_{Lambertian}$$

Also, since this evaluation concerned the external light extraction characteristics of the detachable film, the viewing angle and EQE characteristics of each detachable external light extraction film were analyzed at a current density of 1 mA/cm^2^ on the same device.

## 3. Results

To evaluate the optical property of the QD layers, we measured the photoluminescence (PL) according to the spin-coated speed in Figure 3. The graph of PL intensity showed that the PL of the QD/PDMS composite layer was emitted at 660 nm wavelength, and the intensity of generated red light increased as the layer became thicker. Faint blue light leakage still remained.

**Figure 3 micromachines-16-01367-f003:**
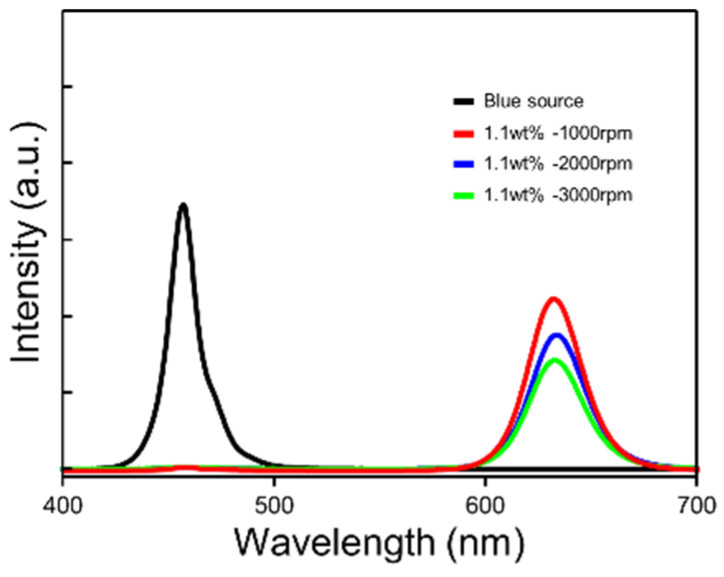
PL spectra of QD/PDMS composite film according to the spin-coating speed.

Figure 4 shows the surface morphology of the PDMS surface based on RIE processing under various gas and ratio conditions. Figure 4a,b show SEM images of a flat PDMS single film and a QD/PDMS composite film before RIE processing, respectively. Since the RIE process had not yet been performed, the surface had no change in morphology and that extended to any light extraction structures on the PDMS surface and QD/PDMS composite film. Figure 4c–e show the surface morphology of the QD/PDMS composite film after RIE processing under different gas and gas ratio conditions. The results were that they caused discrepancies in micro-wrinkle structures. In particular, 4:1 mixed CF_4_ + O_2_ gas led to a higher density of wrinkle and a more distinctive pattern.

Essentially, a unit of PDMS is composed of a Si-O-Si bond, with two methyl groups (CH_3_) bonded to each silicon atom, as silicon atoms can form four chemical bonds with oxygen and two methyl groups. Under the 4:1 mixed CF_4_ + O_2_ gas condition, the methyl groups on the PDMS surface are oxidized and converted into silanol (Si-OH) groups. Oxygen gas including operation gas can be caused by enhancing the roughness of surfaces that incorporate micro-structures, thereby enhancing etching reactivity. Consequently, a structure with a high density of wrinkles is believed to have formed. Meanwhile, as shown in Figure 4c,e, wrinkles were scarcely formed because the primary etching mechanism of PDMS relies on oxidation reactions driven by oxygen radicals. We determined that pure CF_4_ and CHF_3_ plasma lack such oxidative radicals, preventing effective etching of PDMS’s organic components. Furthermore, the hydrogen in CHF_3_ combines with fluorine atoms to make polymer formations. These fluorocarbon polymers were deposited on the PDMS surface, forming a hard protective layer. This layer inhibits the etching reaction of PDMS and interferes with structure formation, resulting in the absence of patterns. These differences in pattern morphology directly impact light scattering and light extraction efficiency when applied externally to OLEDs.

Especially under CF_4_ and CF_4_ + O_2_ conditions, the uniform wrinkle structure is expected to induce both forward and lateral light reflection scattering, contributing to improved viewing angle uniformity and light extraction performance.

Figure 5 shows the improvement in the EQE of OLEDs at a current density of 1 mA/cm^2^ according to RIE conditions. Figure 5a shows results for the QD/PDMS structure, and Figure 5b shows results for the PDMS/QD/PDMS structure. For the QD/PDMS structure, the highest improvement was observed under CF_4_ conditions at 9.2%, followed by CF_4_ + O_2_ at 8.3%, CHF_3_ at 6.6%, and the untreated QD/PDMS film at 7.2%. The EQE enhancement values appeared in the order CF_4_, CF_4_ + O_2_, and CHF_3_, indicating that light extraction efficiency varied depending on the size and dimensions of the surface pattern. Although a more regular wrinkle structure formed under CF_4_ + O_2_ conditions, this shows that the physical size of the pattern had a greater influence on light extraction performance than the pattern’s shape. This result corresponds with previous reports showing that light extraction efficiency was improved by a larger diameter in Micro Lens Arrays (MLAs) [22,23]. On the other hand, −33.5% of the EQE was decreased under the CF_4_ + O_2_ condition compared with the reference device for the PDMS/QD/PDMS structure, and it decreased under all conditions except the CF_4_ condition. This means that light loss and absorption were increased, and light scattering effects were reduced, by the extra PDMS layer. In the case of the multi-layer structure, although the scattering structure became more complex, the interfacial light loss between the QD and PDMS layer increased. Figure 5c,d represent the luminance distribution depending on the viewing angle of single and multiple-layer films. For blue OLEDs, the angular distribution was narrower than the Lambertian range from 0 to 20°. And from 0 to 20°, the single and multiple-layer film expanded the luminescence distribution more than the bare device. Above 20°, the radiation profile was approximal to the Lambertian range. However, in the case of CHF_3_, wrinkles were never formed on the surface, showing an angle distribution almost similar to the untreated RIE condition, and the trend of the EQE was also equal. These results clearly demonstrate the correlation between SEM images and viewing angle characteristics. For the QD/PDMS film, it had similar distributions to the Lambertian range at all viewing angles; the multiple-layer film presented a wider viewing angle distribution than the single-layer film. We determined that the additional PDMS layer formed multiple scattering light paths internally, inducing light to be emitted at wider angles not only from primary scattering at single wrinkles but also from re-scattering between interlayers. These multiple paths caused abnormal scattering directions by light absorption, loss, diffusion, and simple light distribution. In the QDs, multiple light scattering occurs so that the light emission angle light becomes widened and the EQE is reduced, as some of the light is trapped or extinguished in the film. Consequently, we confirmed that the PDMS/QD/PDMS film has an advantage in viewing angles, but there was a trade-off between efficiency and loss paths.

Figure 6a shows the EL spectrum of an OLED with a QD/PDMS composite film applied under different RIE processing conditions. The reference blue OLED exhibits typical blue-light emission characteristics with an emission peak at 471 nm. In contrast, the device with the QD/PDMS composite film clearly showed an emission peak at around 620 nm. This is attributed to the color conversion effect by the red QDs, indicating that the short-wavelength light from the blue OLED was absorbed by the QDs within the film and re-emitted as longer-wavelength red light. The peak intensity around 620 nm in the EL spectrum was stronger in the structure treated with RIE than in the QD/PDMS composite film. Figure 6b shows the CIE 1931 color coordinates calculated based on the emission spectra. The initial color coordinates of the single blue OLED device are located near (x = 0.17, y = 0.38), corresponding to the blue region. However, the color coordinates of all devices with the QD/PDMS composite film shifted toward the red region, changing to the central area (x = 0.28, y = 0.41) on the CIE 1931 diagram, which corresponds to the white light emission region. This shift can be interpreted as a result of both color conversion via the QDs and the light extraction effect of the composite film.

## 4. Discussion

We developed a white light emission device using only a greenish-blue OLED and a red QD external film without a complex multi-EML structure or separate white-light-emitting material. Consequentially, high-efficiency white OLEDs can be fabricated through a simple process. The multi-layer structure exhibited some light loss due to the top PDMS layer, resulting in relatively limited color coordinate shifts. However, color coordinates could be adjusted by controlling surface scattering characteristics based on RIE processing conditions, and it was confirmed that CF_4_-treated QD/PDMS films most effectively achieved white light emission. Unlike conventional multi-stack approaches, this method can be utilized to realize an externally color-converted white OLED with a simple structure that is easily applicable to flexible substrates.

## 5. Conclusions

This study has proven the possibility of high-purity color conversion and light extraction using QD/PDMS composite film with wrinkled structures by using an etching process. The color conversion layer was fabricated by dispersing red QDs in solvent, which was mixed with an uncured PDMS solution and then underwent thermal curing. Subsequently, a micro-wrinkle structure was produced using the RIE process under CF_4_, CF_4_ + O_2_ (4:1), and CHF_3_ gas conditions. The shape and density of wrinkles varied distinctly with different etching gases and mix ratios. Although the wrinkle shape had well-defined features at CF_4_ + O_2_ condition, the highest enhancement occurred under the CF_4_ condition, at 9.2%. These results show that the distribution of wrinkle length and depth is crucial for determining optical scattering length. We determined that the wrinkle structure improves the absorption efficiency of QDs and scattering, thereby enhancing color conversion efficiency and external light extraction efficiency. Our suggested processes have advantages without requiring complex patterning processes or expensive equipment. Additionally, the combination of the greenish-blue OLED and a red-QD-color conversion layer produces white-color OLEDs. We believe this has highly practical applications in display fields. In future work, we expect to expand high-efficiency, high-color-reproducibility display devices by studying various QDs and etching process conditions.

## Figures and Tables

**Figure 1 micromachines-16-01367-f001:**
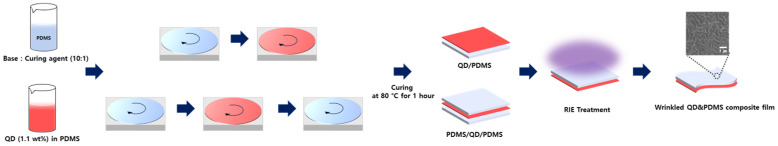
Illustration of QD/PDMS and PDMS/QD/PDMS composite film processing and etching process.

**Figure 2 micromachines-16-01367-f002:**
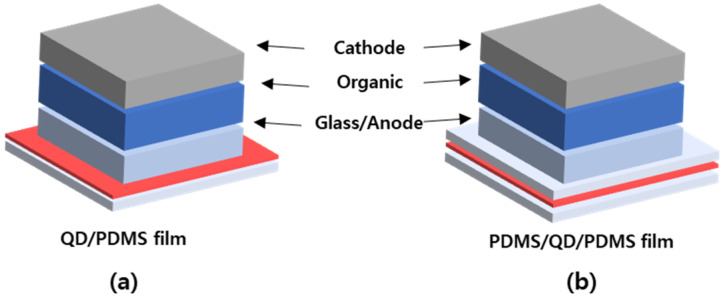
Schematic diagrams present the overall structure utilized with (**a**) QD/PDMS film (single layer) and (**b**) PDMS/QD/PDMS (multiple layers).

**Figure 4 micromachines-16-01367-f004:**
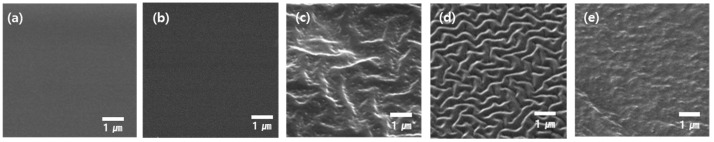
SEM images of morphology (**a**,**b**) before etching process of single flat PDMS and QD/PDMS, after etching process (**c**) CF_4_, (**d**) CF_4_ + O_2_ (4:1), and (**e**) CHF_3_.

**Figure 5 micromachines-16-01367-f005:**
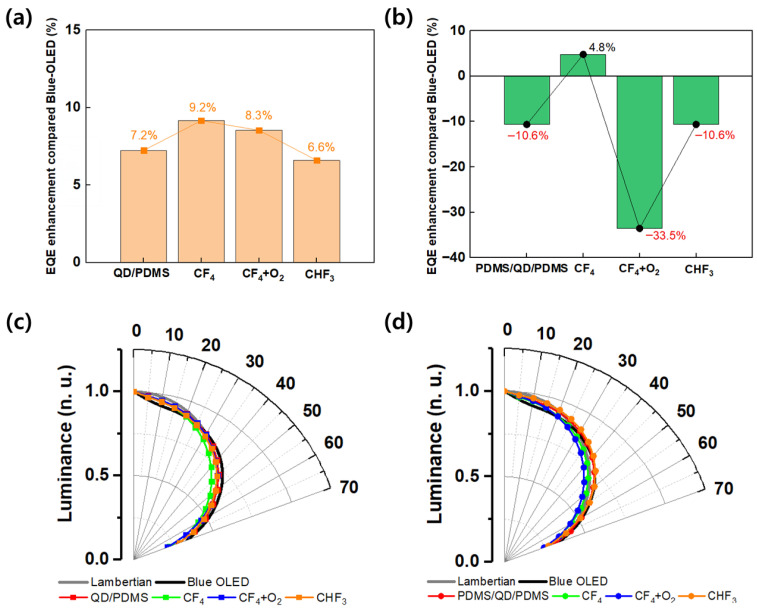
(**a**,**b**) show the EQE enhancement of devices with various RIE processes compared to the reference blue OLED. The viewing angle characteristics of (**c**) QD/PDMS and (**d**) PDMS/QD/PDMS.

**Figure 6 micromachines-16-01367-f006:**
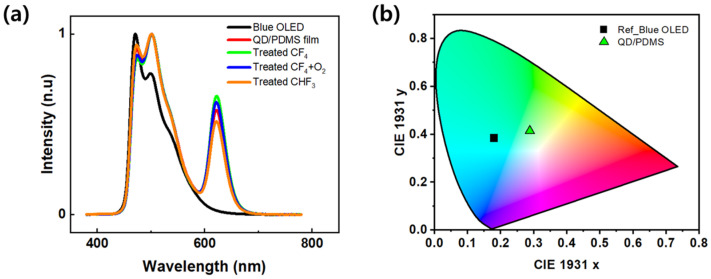
(**a**) EL spectra of blue OLED with QD/PDMS and various RIE-treated films, (**b**) CIE 1931 chromaticity coordinates (inserted square: blue OLED, triangle: QD/PDMS film + blue OLED).

## Data Availability

The original contributions presented in this study are included in the article/Appendix A. Further inquiries can be directed to the corresponding authors.

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
