# Peer review of "Simple Process for Flexible Light-Extracting QD Film and White OLED"

_micromachines, 2025, doi:10.3390/mi16121367_

Round 1

Reviewer 1 Report

Comments and Suggestions for Authors

In their manuscript titled "Simple Processed Flexible Light Extracting QD Film and White OLED", Eun Jeong Bae and colleagues describe a simple and cost-effective method for fabricating a flexible QD-PDMS composite film, which acts as both a color conversion layer and an external light extraction layer for blue OLEDs. The light extraction efficiency is improved by forming micro/nano structures via Reactive Ion Etching (RIE). The work addresses the relevant topic of enhancing light efficiency and color management in flexible displays. However, the manuscript requires significant revisions to address several crucial points related to experimental detail, data interpretation, and, critically, the lack of a detailed understanding of the underlying processes before it can be considered for publication.

Specific Points:

1.  Figure 5: Error bars are missing in (a) and (b). It is unclear how many devices were characterized to obtain these results, and whether the reported values are average values or derived solely from the performance of the best devices. 

2.  Line 178: The authors claim that an anisotropic wrinkle structure is formed. However, based on the provided SEM images in Figure 4, the anisotropy is not obvious. The authors should provide a quantitative characterization to support their claim and a reason why this is the case.

3.  Section 2.3: Crucial experimental parameters are missing such as vacuum pressure maintained during the EL measurements.

4.  In Section 2.1, details for the RIE process used such as the chamber pressure for both the Argon pre-treatment and the subsequent etching steps are missing. 

5.  Line 221: The authors state that the results "clearly demonstrate the correlation between SEM images and viewing angle characteristics". For me, the correlation is not clearly demonstrated and a more sophisticated, quantitative analysis is required to support this claim. 

6.  A concise summary explaining the derivation and significance of the CF​ factor, including its role in correcting for angular emission characteristics should be included in Section 2.3.

Based on all these points, particularly the requirement for a more quantitative, mechanistic discussion linking the RIE gas composition, the resulting anisotropic/isotropic morphology, and the resulting optical performance (EQE/Viewing Angle), I suggest that the paper might be suitable for publication after major revision. 

Comments on the Quality of English Language

A professional proofreading would significantly enhance the overall clarity and readability of the manuscript.

Author Response

Dear Reviewer 1

We have attached answers to your questions.

Please confirm the attached file.

Reviewer 2 Report

Comments and Suggestions for Authors

Please see attached review report.

Author Response

Dear Reviewer 2

We have attached answers to your questions.

Please confirm the attached file.

Reviewer 3 Report

Comments and Suggestions for Authors

This paper describes the preparation of color-conversion films by incorporating quantum dots into a polydimethylsiloxane (PDMS) matrix and the subsequent fabrication of functional devices. Upon revision addressing the following comments, this manuscript is suitable for publication in the journal Micromechanics.

  1. The characterization in the manuscript relies solely on scanning electron microscopy (SEM); incorporating additional analytical techniques would strengthen the study.
  2. The article omits key details regarding the quantum dots used, such as their chemical composition and quantum yield before and after incorporation into PDMS.
  3. While the reactive ion etching (RIE) conditions specify time and power, other essential parameters—such as chamber pressure and gas flow rates—are not provided.
  4. The CIE 1931 chromatic diagram does not clearly delineate the white-light emission region.
  5. The histogram in Figure 5(a) could be enhanced by including reference data from blue OLED devices.

Author Response

Dear Reviewer 3

We have attached answers to your questions.

Please confirm the attached file.

Round 2

Reviewer 1 Report

Comments and Suggestions for Authors

The authors addressed all the points from my first report in sufficient detail. However, this has raised a two new questions.

1. line 119: Can the authors quantify the term "low surface energy"?

2. line 121: The same with "very thin". How thin? 

3. line 230: the authors probably mean "wrinkle", not "winkle"

Comments on the Quality of English Language

A professional proofreading would significantly enhance the overall clarity and readability of the manuscript.

Author Response

Dear reviewer 1

Thank you for your comments.

Q1. line 119: Can the authors quantify the term "low surface energy"?

A1. Due to the methyl group (CH3) attached to the surface of PDMS structure, PDMS surface energy has ranges from 19 ~ 22 mN/m, typically.

Q2. line 121: The same with "very thin". How thin? 

A2. PDMS thickness was 20 um. We modified and it highlighted in manuscript.

Q3. line 230: the authors probably mean "wrinkle", not "winkle"

A3. This is type error. it is modified.

Reviewer 3 Report

Comments and Suggestions for Authors

Accept in present form

Comments on the Quality of English Language

Accept in present form

Author Response

Dear reviewer 2

Thank you for your sincere review. and we have been answered all reviewer's questions.
